# Differential Expression of microRNAs in Serum of Patients with Chronic Painful Polyneuropathy and Healthy Age-Matched Controls

**DOI:** 10.3390/biomedicines11030764

**Published:** 2023-03-02

**Authors:** Antonio Pellegrino, Sophie-Charlotte Fabig, Dilara Kersebaum, Philipp Hüllemann, Ralf Baron, Toralf Roch, Nina Babel, Harald Seitz

**Affiliations:** 1Fraunhofer Institute for Cell Therapy and Immunology, Branch Bioanalytics and Bioprocesses (IZI-BB), 14476 Potsdam, Germany; 2Division of Neurological Pain Research and Therapy, Department of Neurology, University Hospital Schleswig-Holstein, 24105 Kiel, Germany; 3Berlin Institute of Health at Charité—University Clinic Berlin, BIH Center for Regenerative Therapies, 13353 Berlin, Germany; 4Center for Translational Medicine and Immune Diagnostics Laboratory, Medical Department I, Marien Hospital Herne, University Hospital of the Ruhr-University Bochum, 44625 Herne, Germany

**Keywords:** differential expression, microRNA, serum, chronic pain, polyneuropathy

## Abstract

Polyneuropathies (PNP) are the most common type of disorder of the peripheral nervous system in adults. However, information on microRNA expression in PNP is lacking. Following microRNA sequencing, we compared the expression of microRNAs in the serum of patients experiencing chronic painful PNP with healthy age-matched controls. We have been able to identify four microRNAs (*hsa*-miR-3135b, *hsa*-miR-584-5p, *hsa*-miR-12136, and *hsa*-miR-550a-3p) that provide possible molecular links between degenerative processes, blood flow regulation, and signal transduction, that eventually lead to PNP. In addition, these microRNAs are discussed regarding the targeting of proteins that are involved in high blood flow/pressure and neural activity dysregulations/disbalances, presumably resulting in PNP-typical symptoms such as chronical numbness/pain. Within our study, we have identified four microRNAs that may serve as potential novel biomarkers of chronic painful PNP, and that may potentially bear therapeutic implications.

## 1. Introduction

The most prevalent conditions of the peripheral nervous system in adults [1], polyneuropathies (PNP), are a mostly-chronic condition known to be therapeutically challenging, potentially disabling, and even lethal [2,3,4]. Resulting from a lesion or disease of the peripheral nervous system, PNPs become clinically apparent through the so-called stocking-glove pattern [5]: symmetric, initially-distally-distributed sensorimotor signs and symptoms, such as hypoesthesia or hypoalgesia, that may spread further proximally in the course of disease progression. In case of painful PNPs, patients may also experience gain of function, e.g., spontaneous burning sensations or evoked pain.

Interestingly, not all of the PNP patients develop neuropathic pain (NP) [2]. The possible mechanisms leading to a specific clinical phenotype are subject to ongoing investigations, with several hypotheses having been posed so far [1,3,4,5]. What is becoming increasingly clear is that a complex molecular interplay contributes to this process, of which many components seem to be well-orchestrated by microRNAs (miRNAs) [6,7,8].

Non-coding RNAs, called microRNAs, are around 22 nucleotides long and act as transcriptional and post-transcriptional regulators of gene expression [9]. Through binding, deactivation, and/or degradation, they control the messenger RNAs (mRNAs) that they are targeting [10,11]. While numerous distinct miRNAs can bind the same mRNA, a single miRNA often binds to multiple mRNAs [12]. Although miRNAs are mostly found in cells, there are populations that are exported outside: circulating miRNAs. These can be found in various body fluids, including blood, urine, cerebrospinal fluid, saliva, and tears [13,14]. They can be released through active passage, in microvesicles, exosomes, or through being bound to a protein [15,16,17,18]. In addition, they may also be passively released during cell injury, thus potentially reflecting the extension of neural damage. Hence, miRNAs might serve as potential biomarkers of injuries causing acute pain, and as in most cases of PNP, chronic pain. So far, information on miRNA expression in PNP is lacking and only few miRNAs, involved in the process of chronic painful PNP, have recently been examined. For example, miR-132-3p showed a pro-nociceptive effect in peripheral neuropathies with chronic neuropathic pain [6]. In addition, depending on the examined site, miRNAs miR-146a and miR-155 were expressed aberrantly when compared to healthy controls [19]. In another study, miR-223 was associated with the attenuation of neuronal activity in pain pathways [7]. However, these findings were determined via qPCR using, inter alia, highly invasive sural nerve biopsies and nucleus pulposus tissue. Since PNP is currently diagnosed mostly based on characteristic clinical signs and symptoms that are to be validated through further, at best objectifiable examination methods [20], the diagnostic workup can be time-consuming, expensive, often unsuccessful, and requires qualified medical professionals [21]. Thus, a fast, easy, and non-invasive but also accurate, reliable, and objective method of detection is urgently needed.

The aim of our study was to non-invasively determine the entire differential expression of miRNAs in the serum of chronically pain-suffering PNP patients, in order to suggest novel disease biomarkers and novel disease mechanisms, as well as to identify the most likely candidates for novel directions in therapy development.

## 2. Materials and Methods

### 2.1. Patient Selection, Serum Separation and Storage

Patients were recruited at the University Hospital Schleswig-Holstein Campus Kiel (research group around Philipp Hüllemann) and the Marien Hospital Herne (research group around Nina Babel). In total, 30 patients with chronically painful PNP (ø 59 years) as well as 30 age-matched healthy patients (ø 58 years) were selected. Each group contained 10 men and 20 women. The average height in the control group was 175.63 ± 9.59 cm, with an average weight of 72.28 ± 8.83 kg, compared to 177.98 ± 8.80 cm and 85.51 ± 16.87 kg in the patient group. PNP patients were selected with an average pain of ≥ 4 on a numerical scale (0–10), a chronification score on the Mainz Pain Grading System of ≥ II, and a pain duration ≥ 6 months. The PNP etiologies were distributed as follows: chemotherapy-induced: *n* = 3; diabetic: *n* = 3; vitamin deficiency: *n* = 1; chronic inflammatory demyelinating polyneuropathy: *n* = 3; hereditary: *n* = 1; unclear etiology: *n* = 19.

For serum separation, whole blood in a primary blood-collection tube (without clot activator and without anticoagulants) was collected. For complete clotting, tubes were left at room temperature for 30 min and centrifuged for 10 min at 1900× *g* and 4 °C. Subsequently, the serum phase was transferred to a new tube. Serum samples in new tubes were centrifuged again for 15 min at 3000× *g* and 4 °C. After centrifugation, the cleared supernatant was carefully transferred to a new tube. For storage, the separated serum was kept frozen in aliquots at −80 °C. Before processing, room-temperature-thawed serum samples were centrifuged for 5 min at 3000× *g* and 4 °C to remove cryoprecipitates.

### 2.2. miRNA Purification

Purification of cell-free total RNA, primarily miRNA and other small RNA, from the serum was performed using the miRNeasy Serum/Plasma Advanced Kit (QIAGEN, Hilden, Germany). The purification was performed according to the manufacturer’s protocol, with a starting volume of 200 µL serum. The procedure combined guanidine-based lysis of samples, an inhibitor removal centrifugation step, and a silica-membrane-based purification of total RNA. The purified total RNA was then eluted in 20 µL RNase-free water.

### 2.3. miRNA Pre Library Preparation Quality Control

Pre library preparation quality control was performed using the QIAseq miRNA Library QC PCR Panel (QIAGEN, Hilden, Germany). The primary purpose was to control the quality of the isolated RNA in any next-generation sequencing experiment. The addition of the QIAseq miRNA Library QC Spike-Ins during RNA isolation enabled monitoring of the comparability and reproducibility from RNA isolation to sequencing. The quality control was performed according to the manufacturer’s protocol, with 0.5 μL QIAseq miRNA Library QC Spike-Ins per 200 μL serum and 0.5 μL UniSp6 Spike-In per reverse transcription reaction. To avoid contamination, all samples were prepared under sterile conditions. Each assay was transferred into a LightCycler capillary (Roche, Mannheim, Germany), capped, and briefly centrifuged. Analysis of the samples was performed with the LightCycler 2.0 Instrument (Roche, Mannheim, Germany). After conducting the qPCR-based quality control, the data were compared, outlier samples identified, and considered for exclusion in the library preparation.

### 2.4. miRNA Library Preparation

Library preparation was performed using the QIAseq miRNA Library Kit (QIAGEN, Hilden, Germany), enabling unbiased next-generation sequencing of mature miRNAs using the MiSeq instrument (Illumina, Berlin, Germany) for differential-expression analysis of PNP vs. control samples. The library preparation was performed according to the manufacturer’s protocol, with the recommended starting volume of 5 µL total RNA of the RNA eluate, when 200 µL of serum had been processed using the miRNeasy Serum/Plasma Advanced Kit (QIAGEN, Hilden, Germany). The procedure started with the sequentially-adapter ligation to the 3′ and 5′ ends of miRNAs. Subsequently, universal cDNA synthesis with unique molecular index assignment, cDNA cleanup, library amplification, and library cleanup were performed. The cleaned miRNA library was then eluted in 17 µL RNase-free water.

### 2.5. Adapter Dimer Removal and miRNA Library Pre Sequencing Quantification/Quality Control

Adapter-dimer removal was performed using the BluePippin instrument (Sage Science, Beverly, MA, USA) with 3% agarose cassettes and internal standards, to automatically separate the miRNA sequencing library from their adapter dimers. The BluePippin optical system was calibrated and the continuity test was completed before every run. The size selection was performed according to the manufacturer’s protocol, with the size-selection mode set to tight 180 bp. After adapter-dimer removal, the cleaned miRNA-sequencing library was subjected to the Bioanalyzer 2100 system (Agilent, Santa Clara, CA, USA) for pre sequencing quantification/quality control. Therefore, 1 µL of each miRNA sequencing library was analyzed using a High Sensitivity DNA chip according to the manufacturer’s instructions.

### 2.6. Next Generation Sequencing

The Illumina MiSeq instrument (Illumina, Berlin, Germany) was used to enable next-generation sequencing of mature miRNAs for differential-expression analysis of PNP vs. control samples. Therefore, quality-controlled miRNA-sequencing libraries were diluted to 0.5 nM, and four samples at a time were pooled according to the Illuminas Index Adapter Pooling Guide, considering the color balance. Subsequently, pooled 0.5 nM libraries were denatured and diluted to 20 pM, analogously to the NextSeq System Denature and Dilute Libraries Guide performing the Standard Normalization Method. Denatured and diluted library pools were loaded onto a MiSeq Reagent Kit v3 cartridge with 13 pM and 1% phiX, according to the MiSeq System Denature and Dilute Libraries Guide and performing the Standard Normalization Method from step “Dilute Denatured 20 pM Library”. The sample sheet, the flow cell, the PR2 bottle, the waste bottle, as well as the reagent cartridge were prepared analogously to the MiSeq System Guide. For sequencing, FASTQ Only and TruSeq Small RNA with a 75 bp single reading was chosen to include the added unique molecular indices. Resulting FASTQ files were analyzed with a CLC Genomics Workbench 22 (QIAGEN, Hilden, Germany) and the Biomedical Genomics Analysis plugin 22.0.4 (QIAGEN, Hilden, Germany).

### 2.7. Sample-to-Sample Correlation

The comprehensive set of QIAseq miRNA Library QC Spike-Ins allowed thorough quality control of the NGS data by assessing the reproducibility and linearity of the mapped reads. The 52 QIAseq miRNA Library QC Spike-Ins are synthetic 5′-phosphorylated miRNAs of plant origin and bear no significant homology to human miRNAs. Following mapping of the QIAseq miRNA Library QC Spike-In reads, they were normalized to the total number of reads per sample. After this normalization to individual sample reads was done for all spike-ins in all samples, they were evaluated for normality via Shapiro-Wilks Normality Test, and a Spearman Correlation matrix was plotted for sample-to-sample correlation using Prism 9.1.2 (GraphPad, Boston, MA, USA).

### 2.8. miRNA Differential Expression Analysis, GO Analysis and miRDB Target Prediction

CLC Genomics Workbench 22 (QIAGEN, Hilden, Germany) and the Biomedical Genomics Analysis plugin 22.0.4 (QIAGEN, Hilden, Germany) were used to analyze the FASTQ files of each miRNA library, and to perform a differential expression analysis. For quantification of the miRNA libraries, the QIAseq miRNA Quantification workflow, with default settings, was utilized to annotate the miRNA reads using miRbase v22. For differential expression analysis of the miRNA libraries, the QIAseq miRNA Differential Expression workflow was used with the following settings: “Expression tables: Grouped on mature; Test differential expression due to: Group; While controlling for: Age; Comparisons: Against control group; Control group: Control”. In addition, a minimal reading count of 5 was used for the analysis. Significant results included only those differentially expressed miRNAs with a Bonferroni corrected *p*-value ≤ 0.05 and a fold-change FC ≥ 2.0.

The GO enrichment analysis (biological process) was performed on targets of all differentially-expressed miRNAs identified with the following settings: GO annotation table: goa_human_rna_20181212; exclude computationally inferred GO terms; allow gene name synonyms; ignore gene name capitalization and ignore features with mean RPKM below: 5.0. As GO enrichment analyses with a Bonferroni corrected *p*-value, FDR corrected *p*-value or *p*-value ≤ 0.05, and a fold-change FC ≥ 2.0 were not successful, GO enrichment analysis parameter were changed to *p*-value ≤ 0.05 and a fold-change FC ≥ 1.5. Significant results included only those GO terms with a *p*-value ≤ 0.05 and differentially expressed genes DE > 2.

The miRNA target prediction was performed using miRDB human (Version: 6.0, Prediction Tool: MirTarget V4, miRNA Source: miRBase 22); miRDB is an online database for miRNA target prediction and functional mapping. All the targets in miRDB are predicted by MirTarget, which was developed by analyzing thousands of miRNA-target interactions. From these interactions, common features associated with miRNA binding/target downregulation have been identified and used to predict miRNA targets using machine learning methods [22]. Therefore, the differentially-expressed and filtered miRNAs were subjected to miRDB, and results included only the top 3 targets for each miRNA.

## 3. Results

### 3.1. Patients and Controls

We obtained serum from 30 patients with chronic painful PNP and 30 age-matched control subjects. Each group contained 10 men and 20 women. The average height in the control group was 175.63 ± 9.59 cm, with an average weight of 72.28 ± 8.83 kg compared to. 177.98 ± 8.80 cm with 85.51 ± 16.87 kg in the patient group. The average age of patients vs. controls was 59 vs. 58 years. The following are the PNP etiologies of the patients: chemotherapy-induced: 3; diabetes: 3; vitamin deficiency: 1; chronic inflammatory demyelinating polyneuropathy: 3; hereditary: 1; the remaining etiologies are unknown. In terms of the heterogeneity of etiologies in our cohort, it must be considered that the modern approach to neuropathic pain is to follow a mechanism-based classification and to aim at individualized treatments rather than dividing patients by etiologies [4]. Thus, the aim of our study was to identify biomarkers for neuropathic pain in general without targeting a specific etiology.

### 3.2. Next Generation Sequencing

Prior to library preparation, samples were quality controlled using a miRNA library QC PCR panel. The qPCR assay provided insight into RNA isolation efficiency, cDNA synthesis efficiency, and controlled for endogenous miRNAs as well as for hemolysis. Since all quality controls of all samples were successful, library preparation was performed. After library preparation and ahead of next generation sequencing, samples were cleaned of adapter dimers by automated size selection and quantified/quality controlled. Since the quantification/quality control of all samples was successful, next generation sequencing was performed. Therefore, FASTQ Only and TruSeq Small RNA with a 75 bp single read was chosen using the MiSeq instrument, with 4 samples per flow cell. Resulting FASTQ files were analyzed using CLC Genomics Workbench 22 (QIAGEN, Hilden, Germany) and the Biomedical Genomics Analysis plugin 22.0.4 (QIAGEN, Hilden, Germany). Hence, the reads were annotated with the miRNA quantification workflow with default settings using miRbase v22b. On average, 6.5 M reads (1.5 M unique molecular index grouped reads) were generated per sample and 45.73% of these reads could be annotated.

### 3.3. Sample-to-Sample Correlation

As an additional quality control, all samples were spiked with 52 miRNA library QC spike-ins before miRNA isolation from serum. These spike-ins are artificial, 5′-phosphorylated, plant-derived miRNAs, and they are not significantly homologous to human miRNAs [23]. The miRNA library QC spike-in reads were mapped, and their counts were normalized to total reads per sample. All spike-ins in all samples were normalized to individual sample reads. Subsequently, they were evaluated for normality after which a correlation matrix was plotted to allow sample-to-sample correlation. For all samples, the spearman r values ranged from 0.86–0.99, showing strong correlation through all sample preparations (Figure 1).

### 3.4. Differential Expression

For differential expression analysis, the resulting quantification of each miRNA library was utilized and grouped based on being PNP or control. Although the groups were age-matched, differential expression analysis took age into account to exclude age-based false-positive results. In addition, a minimal reading count of 5 was used for the analysis. Only differentially expressed miRNAs with a fold-change FC ≥ 2.0 and a Bonferroni adjusted *p*-value ≤ 0.05 were included in the significant results overview, which is given by Table 1.

The *hsa*-miR-3135b was determined with the largest fold change of -6.30 and a highly significant Bonferroni corrected *p*-value of 5.94 × 10^−11^. Two other miRNAs were downregulated as well: *hsa*-miR-584-5p with the smallest fold change of −2.62 and a Bonferroni corrected *p*-value of 1.43 × 10^−6^, as well as *hsa*-miR-12136 with a fold change of −3.80 and a Bonferroni corrected *p*-value of 7.50 × 10^−4^. The only upregulated miRNA was *hsa*-miR-550a-3p with a fold change of 4.27 and a significant Bonferroni corrected *p*-value of 0.02, respectively.

### 3.5. GO Enrichment Analysis and miRNA Target Prediction of Differentially Expressed miRNAs

Subsequently, a GO enrichment analysis (biological process) was performed on targets of all differentially-expressed miRNAs identified. As GO enrichment analyses with a Bonferroni corrected *p*-value, FDR corrected *p*-value or *p*-value ≤ 0.05, and a fold-change FC ≥ 2.0 were not successful, the following GO enrichment analysis was performed with a *p*-value ≤ 0.05 and a fold-change FC ≥ 1.5. The summary of the GO enrichment analysis is given by Table 2.

GO biological-processes analysis included 13 terms. Among them, vascular endothelial growth factor/vasculature development/angiogenesis pathways, also signal transduction/cell communication pathways and cell apoptotic process/developmental process related pathways, indicated that blood vessels and their surrounding tissues, such as smooth muscle cells and neurons, are actively degenerating concurrently with PNP progression. In-depth miRNA target prediction supported these findings by determining targeted genes/mRNAs/proteins. This target prediction was performed using miRDB’s human target prediction (Version: 6.0, Prediction Tool: MirTarget V4, miRNA Source: miRBase 22) [22]. The summary of the miRNA target prediction is given in Table 3 and shows the top three predictions for each differentially expressed and filtered miRNA:

The *hsa*-miR-3135b target prediction revealed the gene “leucine rich repeat containing 27” (*LRRC27*) as a potential target. *LRRC27* is expressed in platelets and its overexpression is associated with preeclampsia, which, among other symptoms, manifests in high blood pressure [24]. In addition, the gene “formin like 3” (*FMNL3*) is assigned to *hsa*-miR-3135b. *FMNL3* is expressed in endothelial cells and is a known cytoskeletal regulator of angiogenesis [25]. Finally, the gene “tetratricopeptide repeat domain 21B” (*TTC21B*) is potentially targeted by *hsa*-miR-3135b. *TTC21B* is expressed in the primary cilium and some variants are associated with arterial hypertension [26].

The *hsa*-miR-584-5p target prediction revealed the gene “USP6 N-terminal like” (*USP6NL*) as a potential target. *USP6NL* is a GTPase-activating protein that functions as a deubiquitinating enzyme, regulating endocytosis and signal transduction [27]. Furthermore, the gene “arginine vasopressin receptor 1A” (*AVPR1A*) is assigned to *hsa*-miR-584-5p. *AVPR1A* is expressed in peripheral blood vessels, encoding a receptor for arginine vasopressin that helps blood vessels to constrict and control blood pressure [28]. Eventually, the gene “SET domain containing 5” (*SETD5*) is potentially targeted by *hsa*-miR-584-5p. *SETD5* seems to control neural cell proliferation and synaptic activity/connectivity [29,30].

The *hsa*-miR-12136 target prediction revealed the gene “immunoglobulin superfamily member 11” (*IGSF11*) as a potential target. *IGSF11* is required for synaptic development [31]. In addition, the gene disheveled associated activator of morphogenesis 1 (*DAAM1*) is assigned to *hsa*-miR-12136. *DAAM1* has been shown to aid in the development of neuronal systems, through nucleating, elongating, and possibly bundling actin [32,33]. Finally, the gene “syntaxin binding protein 5” (*STXBP5*) is potentially targeted by *hsa*-miR-12136. *STXBP5* is expressed in human endothelial cells and is associated with venous thromboembolism that manifests with blood clots, and leads to secondary changes in the blood vessels and alterations to the blood flow [34].

The *hsa*-miR-550a-3p target prediction revealed the gene “myosin heavy chain 10” (*MYH10*) as a potential target. It has been found that *MYH10* participates in the critical developmental process of coronary-vessel formation, and that *MYH10* is able to interact with sodium channels, modulating their current density and gating properties [35,36]. Furthermore, the gene “synaptotagmin 4” (*SYT4*) is assigned to *hsa*-miR-550a-3p. *SYT4* regulates membrane traffic in neurons and seems to be important for dendrite growth [37]. Eventually, the gene “myelin transcription factor 1 like” (*MYT1L*) is potentially targeted by *hsa*-miR-550a-3p. *MYT1L* promotes axonal development/differentiation, neurite outgrowth/proliferation, synaptic transmission, extracellular matrix composition, as well as remyelination after induced demyelination [38,39,40].

## 4. Discussion

Under the chosen criteria, our analysis of the differential expression of miRNAs revealed a significant difference for four annotated miRNAs in the serum of chronic painful PNP patients compared to healthy, age-matched controls. Considering that this number represents only a relatively small percentage, these findings appear to be indicative of possible biomarkers with high accessibility, also possibly-high specificity and sensitivity, for the diagnosis of chronic painful PNP. The identification of the hereby-presented miRNAs may be the first step towards identifying unknown disease mechanisms and towards eventually developing innovative therapeutic approaches. The differentially-expressed miRNAs’ GO biological-processes analysis points to their potential participation in signal transduction, blood-flow regulation, and degenerative processes. A potential diagnostic role of the hereby-identified miRNAs is supported by our findings that the differentially expressed miRNAs identified in the serum of PNP patients were linked to genes/mRNAs/proteins via miRDB that could potentially trigger PNP (and explain its related symptoms).

Confirming the potential targets and their effects/pathways found by our miRDB and the GO analysis, our literature research yielded the first hints of their involvement in hemorheological abnormalities such as hypertension/coronary artery calcification (*hsa*-miR-3135b), pulmonary arterial hypertension (*hsa*-miR-584-5p), angiogenesis/hypoxia (*hsa*-miR-12136), and damaged vascular smooth muscles (*hsa*-miR-550a-3p) [41,42,43,44,45,46]. Since the control group was lighter than the patient group (72.28 ± 8.83 kg vs. 85.51 ± 16.87 kg), the *hsa*-miR-584-5p and *hsa*-miR-550a-3p results may be due to this difference, as both miRNAs have also been linked to obesity/insulin resistance [47,48]. However, logistic regression analyses showed that the differences seen between the control group and the patient group were not due to difference in body weight (Appendix A). Nevertheless, extensive manual literature research via PubMed/PubMed Central, using search terms such as “miRNA polyneuropathy” and “miRNA”, where miRNA is the corresponding differentially expressed miRNA, did not yield results to verify the determined miRNAs in the context of PNP or chronic pain [49]. Thus, to our best knowledge, our data provide further research targets for PNP and chronic pain that might possibly serve as novel disease biomarkers. Eventually, this might help develop new therapeutic options in the future.

These results lead to the hypothesis that a combination of high blood flow/pressure and neural activity dysregulations/imbalances could lead to chronic painful PNP. According to our data, one could speculate that vasoconstriction/high blood pressure induced by upregulation of LRRC27, AVPR1A, TTC21B, STXBP5, downregulated coronary vessel formation (MYH10), as well as overactive angiogenesis/EGFR (FMNL3/USP6NL), lead to the degeneration of small sensory neurons due to the lack of nourishment/restricted blood flow [24,25,26,27,28,34,35]. In addition, neural dysregulation in terms of overactive synaptic development/proliferation and activity/connectivity mediated by upregulation of SETD5, IGSF11, and DAAM1, as well as downregulated dendrite extension/maintenance of neuronal identity (SYT4/MYT1L), could contribute to the degeneration of sensory neurons as well [30,31,32,37,40]. Altogether, overactive synaptic development/proliferation and activity/connectivity, as well as the lack of nourishment/restricted blood flow due to vasoconstriction/high blood pressure, could lead to the degeneration of small sensory neurons resulting in the neuropathy’s typical symptoms such as numbness/pain.

In line with our findings, in a review examining the association of vascular changes with clinical symptoms in animal and human models, the authors identified metabolic changes accompanied by vascular dysfunction as a possible cause (and therapeutic target) of diabetic neuropathy [50]. As for the mechanism, vasoconstriction/high blood pressure is reported to result in reduced nerve blood flow/nerve hypoxia and presumably neuropathy, since the degree of neuropathy was shown to be closely correlated with the number of closed vessels [51].

Perhaps surprisingly, both adult neurogenesis and neuroplasticity have been reported to contribute to the maintenance of long-lasting NP (i.e., chronicity) [52] and NP that may occur adjacent to or remotely from any injury site [53].

## 5. Conclusions

Taken together, the strong correlation between the reported miRNAs (i.e., *hsa*-miR-3135b, *hsa*-miR-584-5p, *hsa*-miR-12136 and *hsa*-miR-550a-3p) and chronic painful PNP is promisingly suggestive of a possible diagnostic and therapeutic usefulness of these biomarkers in the future. In addition, the GO enrichment analysis and the miRNA target prediction suggest possible molecular links between degenerative processes, blood flow regulation, and signal transduction, that might eventually cause PNP and presumably result in typical symptoms such as chronic numbness/pain. However, a qRT-PCR validation would have been desirable but the serum samples were completely used for NGS, and therefore qRT-PCR validation was not possible. Nevertheless, we see this as an exploratory study in which the first step has been taken by sequencing. In a second step, these exploratory results will be confirmed with a larger cohort by qRT-PCR, as the present study is part of a larger project on which we are currently working, in order to shed some light on the complex underlying mechanisms of chronic painful PNP.

## Figures and Tables

**Figure 1 biomedicines-11-00764-f001:**
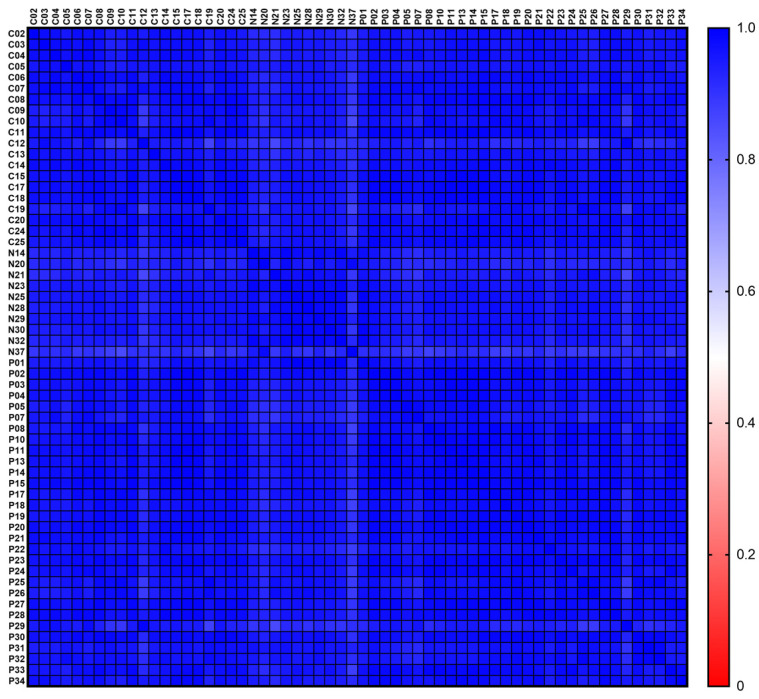
Normalized sample-to-sample correlation matrix. The spearman *r* value legend ranges from blue (1.0) over white (0.5) to red (0.0). For all samples the spearman r values ranged from 0.86–0.99, indicating strong correlation.

**Table 1 biomedicines-11-00764-t001:** Differential expression analysis. Differentially expressed miRNAs (*hsa*-miR-3135b, *hsa*-miR-584-5p, *hsa*-miR-12136, and *hsa*-miR-550a-3p) with a Bonferroni corrected *p*-value ≤ 0.05 and fold-change FC ≥ 2.0.

Name	Log_2_ Fold Change	Fold Change	*p*-Value	FDR *p*-Value	Bonferroni
*hsa*-miR-3135b	−2.65	−6.30	3.53 × 10^−14^	2.26 × 10^−11^	5.94 × 10^−11^
*hsa*-miR-584-5p	−1.39	−2.62	8.50 × 10^−10^	2.72 × 10^−7^	1.43 × 10^−6^
*hsa*-miR-12136	−1.92	−3.80	4.46 × 10^−7^	9.51 × 10^−5^	7.50 × 10^−4^
*hsa*-miR-550a-3p	2.09	4.27	1.45 × 10^−5^	2.07 × 10^−3^	0.02

**Table 2 biomedicines-11-00764-t002:** GO enrichment analysis. GO enrichment analysis (biological process) of all differentially expressed miRNAs. Significant results include only differentially expressed genes DE > 2 with a *p*-value ≤ 0.05.

GO Term	Description	DE Genes	*p*-Values
0030947	regulation of vascular endothelial growth factor receptor signaling pathway	4	1.90 × 10^−4^
0030949	positive regulation of vascular endothelial growth factor receptor signaling pathway	4	1.90 × 10^−4^
1904018	positive regulation of vasculature development	10	4.62 × 10^−3^
0045766	positive regulation of angiogenesis	9	9.00 × 10^−3^
0009967	positive regulation of signal transduction	9	0.01
0010647	positive regulation of cell communication	9	0.01
0023056	positive regulation of signaling	9	0.01
0051094	positive regulation of developmental process	13	0.02
0010656	negative regulation of muscle cell apoptotic process	5	0.02
0090287	regulation of cellular response to growth factor stimulus	6	0.02
0010664	negative regulation of striated muscle cell apoptotic process	4	0.02
0010667	negative regulation of cardiac muscle cell apoptotic process	4	0.02
0090050	positive regulation of cell migration involved in sprouting angiogenesis	5	0.02

**Table 3 biomedicines-11-00764-t003:** miRNA target prediction using miRDB. Top 3 predictions for each differentially expressed miRNA (*hsa*-miR-3135b, *hsa*-miR-584-5p, *hsa*-miR-12136, and *hsa*-miR-550a-3p). *hsa*-miR-3135b targets *LRRC27*, *FMNL3*, and *TTC21B*; *hsa*-miR-584-5p targets *USP6NL*, *AVPR1A* and *SETD5*; *hsa*-miR-12136 targets *IGSF11*, *DAAM1* and *STXBP5*; *hsa*-miR-550a-3p targets *MYH10*, *SYT4* and *MYT1L*.

miRNA	Target Rank	Target Score	Gene Symbol	Gene Description
*hsa*-miR-3135b	1	99	*LRRC27*	leucine rich repeat containing 27
	2	96	*FMNL3*	formin like 3
	3	96	*TTC21B*	tetratricopeptide repeat domain 21B
*hsa*-miR-584-5p	1	96	*USP6NL*	USP6 N-terminal like
	2	95	*AVPR1A*	arginine vasopressin receptor 1A
	3	94	*SETD5*	SET domain containing 5
*hsa*-miR-12136	1	100	*IGSF11*	immunoglobulin superfamilymember 11
	2	100	*DAAM1*	dishevelled associated activator of morphogenesis 1
	3	100	*STXBP5*	syntaxin binding protein 5
*hsa*-miR-550a-3p	1	98	*MYH10*	myosin heavy chain 10
	2	98	*SYT4*	synaptotagmin 4
	3	95	*MYT1L*	myelin transcription factor 1 like

## Data Availability

The data presented in this study are available on request from the corresponding author.

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
