# Peer review of "Differential Expression of microRNAs in Serum of Patients with Chronic Painful Polyneuropathy and Healthy Age-Matched Controls"

_biomedicines, 2023, doi:10.3390/biomedicines11030764_

Round 1

Reviewer 1 Report

Pellegrino et al. studied the differential expression of miRNAs in serum of patients with chronic painful polyneuropathy and healthy age-matched controls ( a cohort of 30 patients and 30 controls) through next sequencing sequencing. The premise of the work is very interesting and the overall structure is well presented. I have some comments:

The abstract is extensive, a suggestion could be the elimination of the first part, it’s like a review. The authors could directly stress the important points of their study

“Materials and Methods” section is descriptive enough. Perhaps a workflow of the procedure could be helpful and included.

Regarding gene ontology, any evidence about cellular component and molecular function?

I wonder if authors would provide a short paragraph or conclusion summarizing some major points, challenges and/ or answers of some demanding questions of the discussed area. I believe it would further improve the scientific value of the paper.

Reviewer 2 Report

I have read with difficulties the study by Pellegrino et al. abour differential expression of serum miRNAs between patients with chronic painfull polyneuropathy and age-matched healthy controls. Manuscript contains a lot of English missuse and as is, it must go throught the very rigorous English proof-reading to make manuscript easier to read. Besides the English, study design is innapropriatte and inapropriately described (as specified in my comments bellow – authors do not define the patient and control cohort well, in their study they do not confirm NGS results by qRT-PCR etc.). All over the manuscript lack scientific soundess and contains numerous missleading and inaccurate sentences. Specific suggestion and comments to potentially improve the manuscript are provided bellow.

1)      Abstract is very difficult to read, there are abundant words and misleading sentences. Native English proof-reading shall be performed.

2)      Introduction contains several inaccuracies, e.g. „populations that migrate outside“ (line 59) is not scientifically sounding (as miRNAs are exported from the cells…they do not migrate…). Sentence „In addition, they may also be passively released during cell injury, thereby potentially serving as biomarkers of injuries causing acute pain or chronic pain“ shall be rephrased to make more sense (e.g. In addition, they may also be passively released during cell injury, reflecting potentially the extension of  neural damage“). Last but not the least, authors state that in the field of PNP only 2 miRNAs were described, but there are numerous studies focusing on the roles of miRNAs in PNPs (e.g. https://pubmed.ncbi.nlm.nih.gov/32454150/, https://pubmed.ncbi.nlm.nih.gov/27836180/ + many others) and even the same authors that are cited in the submitted manuscript (Leinders et al, citation 11) published data about other miRNAs involved in chronic painful PNPs (https://pubmed.ncbi.nlm.nih.gov/28870579/ ). Thus the introduction provided by the authors is insufficient.

3)      Methods lines 172 – 174, i.e. text „… Differential Expression workflow was used with the following settings: Expression tables: Grouped on mature, Test differential expression due to: Group, While controlling for: Age, Comparisons: Against control group and Control group: Control“ – this is just some series of words that makes no sense. Please, edit and explain to the readers what is the meaning of this subsection.

4)      Methods, lines 175-176 authors state that significant results were considered by the fold change  2 – did authors also take into account the total number of reads is all samples? Ie. depth of the sequencing? From our experience, minimum number of reads shall be defined (e.g. 50), as during the statistical processing you may get intereting results (e.g. FC above 10), however, this results is based only on comparison from 3 vs. 3 (from you 30 vs 30) samples as in other samples (27 vs 27) miRNAs were not sequenced at all. Did the authors set up a minimal number of reads prior the analysis?

5)      Methods: Authors performed only NGS, but they did not validate their data using qRT-PCR. As a part of every exploratory study like the one provided by the authors, at least technical validation must be performed on the same samples using qRT-PCR to confirm, that identified miRNAs are indeed present in the samples and that the results is not „false possitive“. Furthermore, independent cohort shall be involved to definitelly confirm observed changes from the exploratory cohort, as NGS studies on different populations are known to provide very incoherent and contradictory results (one can say that „everyone who did NGS identifies his/her own miRNA profile for specific diasese in specifis population – that is why vaidation is neccesarry). So, at least technical validation on the same patient cohort shall be performed to support the results, with independet cohort validation to realy confirm the results being much anticipated.

6)      Lines 195 -  200: Patient information is totally insufficient for scientific publication and for the „exploratory study“ like this. Authors performed miRNA profiling on 30 indiviudal with „polyneuropathy“, that was only defined by severity. But there is nothing about aetiology, which may totally alter miRNA levels (e.g. differences between diabetic PNP or toxonutriive PNP or traumatic PNP would affect miRNA levels, thus not specifying used cohort is useless).  Also patients comorbidites must be metioned as most of them will affect miRNA levels. Also more specifically the healthy controls must be defined and they shall not be matched only by age, but also by BMI and comorbidities (e.g. comparing patients with diabetic PNP versus healthy age-matched individuals does not mean that observed differences are caused by PNP –thay may be caused by diabetes etc.).

7)      Lines 202 – 227 are not results, but methods. Authors also repeat themselves (repeating same sentences from methods). There is an typo – paragraph is named “s-to-s comparison“ and figure is named „s-to-s correlation“. Also Figure 1 is not a result „per se“ – it is just authors justification, that quality control worked – which shall be mentioned in methods,not in results; Figure 1 itself can be totally omitted as it provides no additive value than just stating „For all samples, the spearman r values ranged from 0.86-0.99, showing strong correlation“.

8)      Figure 2,3 and 4 are Tables…thus I would preffer them to be named and numbered as tables.

9)      Results, lines 284 – 320: these are not the results of your study, this is discussion, or maybe not even the discussion but just the list of potential roles of identified target of yours identified miRNAs. Within these paragraphs you are repeating the same strucutre of sentences and much more suitable it would be to redo these paragraphs into the table (e.g. make it a part of Figure 4). Furthermore, for AVPR1A you do not mentioned its function at all (only where it is expressed), moreover, some statements are misleading, e.g. „STXBP5 is expressed in human endothelial cells and is associated with venous thromboembolism that manifests in impaired blood flow, vessel walls and coagulation factors“ – how do you mean that venous thrombemolism manifests in impaired blood flow of „impared vessel walls“? these are clinically nonsenses – venous thrombembolism manifest with blood clots that originate from different reasons and leads to secondary changes in the vessels due and alter blood flow.

10)   Line 323 „to be significantly differentially expressed in the serum of chronic painful PNP patient“ shall be finihes by „… compared to healthy age-matched controls“

11)   Discussion – sentence „These results lead to the conclusion that a combination of high blood flow/pressure and neural activity dysregulations/disbalances are two of the various mechanisms leading to chronic painful PNP.“ is non true. Authors did not perform the pathophysiological and clinical study to confirm that blood pressure/flow or neural activity dysregulations are two mechanisms leading to PNP – results of the current study is that authors identified 4 serum miRNAs differentially expressed between PNP patients and age-matched controls using nGS. All other „results“ descrbied by the authors are just speculations and hypotheses based on literature search.

12)   Conclussion section with conluding remarks about the whole paper is missing (last paragraph of the discussion is not sufficiently summarizing the results of the study).

Round 2

Reviewer 2 Report

I have read with the greatest interest the revised version of the manuscript by Pellegrino et al. English was tremendously improved, which resulted in the huge improvement in the quality of the manuscript that is now easier to read, the ideas in it are more clearly presented and it shows greated inner coherence. Nevertheless, I still have reasons (see bellow) to ask authors to perform technical validation of their RNA-seq data and moreover, after providing information about aetiologies of PNPs in their patients, I request other information. My specific comments can be found bellow togerther with a few identified typos.

Major comments:

1.     Authors state in their answers that „There is no reason to doubt the sequencing results“ – I do not want authors to think that I am accusing them from research misconduction, but there is still a huge discussion about this topic in the scientific community and according to the current opinions, authors shall provide technical validation of NGS data using qRT-PCR. If miRNA profiling is performed using PCR-Arrays, there is no need to validate the data, however, qRT-PCR still remains the gold standard for miRNA quantification. The „era when performing RNA-sequencing and publishing the results was enough“ is slowly over and again, from our experience, even if the experiment is properly set up, you perform proper statistics and get 4-10 miRNAs from RNA-sequencing, qRT-PCR validation usually works only in 1-2 miRNA species. If the identified miRNAs by the authors shall serve as novel biomarker, they will be determined in the clinical practice more likely using qRT-PCR and not NGS, thus at least technical validation of obtained data shall be provided.

2.     Patients description is still insuficcient. Now we only know the aetiology of the PNP, but we have no idea how many men x women were included in each group (e.g. „the worst scenario“ - were there 30 healthy women compared to 30 PNP men?), we have no idea whether PNP patients were higher/heavier compared to controls (i.e. height, weight, BMI).

3.     Considering the added information about etiologies: including so many various aetiologies of PNP into pilot exploration study is a methodological mistake – for exploration study using NGS, patient cohort must be strictly defined to obtain valid results and to be sure that the observed changes in miRNA levels are caused by/related to the studied disease. By including so many aetiologies, it is more probable that the identified miRNAs are associated to the pain itself, than to the specific PNP underlying it. Can authors comment on this?

4.     In the discussion autohrs state (line 369): „possible biomarkers with high accessibility, specificity and sensitivity for the diagnosis of chronic painful PNP“ – could autohrs provide the values of sensitivity and specifity and cut-off values for identified miRNAs to support this statement?

Minor comments:

5.      Abstract, last sentence is unreadable due to copy/paste/editing errors and shall be edited , e.g."Within our study we have identified several microRNAs that may serve as potential novel biomarkers of chronic painful PNP and potentially bear therapeutic implications.

6.     I still believe, as in my previous report, that Figure 1 is abundant (can be provided as supplementary materials) and the related section belongs to the methods, not to the results.

Typos:

·       Line 39: „condition“ shall be „conditions“

·       Line 250: missing dot „serum These

·       Line 372: meachnisms

Round 3

Reviewer 2 Report

Dear authors.

I have read once again your revised version, which I am grateful for. You made a tremendous improvements in the manuscript from the moment I saw its first draft. Yet, there are still some inaccuracies that shall be edited and that you can find bellow.

1) Thank you for all your explanations you provided to me, I understand that you are currently lacking the biological material to perform the technical validation using qRT-PCR. Please, this shall be mentioned as the limitation of the study. I undestand that this was the exploratory study and that all the material was used - this may happed, so, please, state this in the manuscript in the limitations.

2) Considering the different aetiologies - please, also edit the manuscript the way that it is clearer that you were looking for markers of pain than the markers of PNP itself. Your explanation you provided to me is brilliant, so please, inform the readers about your goal.

3) Thank you for providing more details about the patients - I am happy to see that the groups were age and sex matched, however, when looking on their height and weight information:

a) when providing the values of continuous variables, you shall provide either the average +- standard deviation (in case of normally distributed data, e.g. 185 +- 5cm) or median (interquartile range), eg. (182 (174-190) cm) in case of abnormally distributed data. Can you provide this please?

b) I see that the control group was significantly heavier that the study group (72 vs. 85 kg!). Please, perform the logistic regression of your data to show, that the observed differences in miRNA levels are not caused by the obesity instead of the neuropathy (miR-584 and miR-550 have already been linked to obesity - https://pubmed.ncbi.nlm.nih.gov/26319141/ and https://pubmed.ncbi.nlm.nih.gov/31024340/.
